# Structural insight into the electron transfer pathway of a self-sufficient P450 monooxygenase

Lilan Zhang [1,3], Zhenzhen Xie[1,3], Ziwei Liu[1,3], Shuyu Zhou[1], Lixin Ma [1], Weidong Liu [1], Jian-Wen Huang[1], Tzu-Ping Ko [2], Xiuqin Li[1], Yuechan Hu[1], Jian Min[1], Xuejing Yu[1], Rey-Ting Guo [1✉] & Chun-Chi Chen [1✉]

Cytochrome P450 monooxygenases are versatile heme-thiolate enzymes that catalyze a wide range of reactions. Self-sufficient cytochrome P450 enzymes contain the redox partners in a single polypeptide chain. Here, we present the crystal structure of full-length CYP116B46, a self-sufficient P450. The continuous polypeptide chain comprises three functional domains, which align well with the direction of electrons traveling from FMN to the heme through the [2Fe-2S] cluster. FMN and the [2Fe-2S] cluster are positioned closely, which facilitates efficient electron shuttling. The edge-to-edge straight-line distance between the [2Fe-2S] cluster and heme is approx. 25.3 Å. The role of several residues located between the [2Fe-2S] cluster and heme in the catalytic reaction is probed in mutagenesis experiments. These findings not only provide insights into the intramolecular electron transfer of self-sufficient P450s, but are also of interest for biotechnological applications of self-sufficient P450s.

[1] State Key Laboratory of Biocatalysis and Enzyme Engineering, Hubei Collaborative Innovation Center for Green Transformation of Bio-Resources, Hubei Key Laboratory of Industrial Biotechnology, School of Life Sciences, Hubei University, Wuhan 430062, P. R. China. [2] Institute of Biological Chemistry, Academia Sinica, Taipei 11529, Taiwan. [3]These authors contributed equally: Lilan Zhang, Zhenzhen Xie, Ziwei Liu. ✉email: guoreyting@hubu.edu.cn; ccckate0722@hubu.edu.cn

Cytochrome P450 monooxygenases are a group of heme-thiolate proteins that catalyze reactions related to natural products biosynthesis, carbon source assimilation, cellular component synthesis, and xenobiotic metabolisms[1,2]. The P450 enzymes show remarkable substrate promiscuity and can catalyze an array of challenging oxidative reactions[3]. These characteristics highlight the potentials of P450s for biotechnological applications and pharmaceutical industries[4]. P450s acquire electrons from reduced pyridine nucleotides (NADPH or NADH) to generate iron-oxygen intermediates that in turn oxidizes the substrates[5,6]. Most P450s interact with auxiliary redox partners such as flavoenzymes or iron-sulfur proteins to obtain electrons from NAD(P)H[7,8]. Therefore, a functional P450 system requires co-expression with the cognate or compatible redox partners. In this regard, self-sufficient P450 enzymes that contain the redox partners in a single polypeptide chain are attractive biocatalysts for many biotechnological applications, as there is no need to search for suitable redox partners to establish a functional system. For self-sufficient P450s, understanding the intramolecular electron transfer pathway is of great interest.

Two classes of self-sufficient P450s have been identified. In one class, an N-terminal P450 heme domain is fused to a C-terminal NADPH:cytochrome P450 reductase (CPR) that contains a flavin mononucleotide (FMN)-binding flavodoxin and a FAD/NADPH-binding domain (Supplementary Fig. 1a). The electron transport chain in these P450s is NADPH→FAD→FMN→heme. The best studied self-sufficient CYP102A from *Bacillus megaterium* (P450BM3) belongs to this family[9]. Although no full-length structure is available, the electron transport mechanisms of these P450s have been proposed based on the complex structures of partial fragments of the protein and mammalian CPRs. First, the complex structure of heme domain and flavodoxin of P450BM3 indicates a possible route for electron shuttling from the FMN to the heme[10] (Supplementary Fig. 1b). Notably, the polypeptide chain comprising the heme domain and FMN-binding flavodoxin domain was broken into two individual regions with a 20-residue long linker missing (Supplementary Fig. 1b). In addition,

structural and functional studies of homologous mammalian CPRs indicate that significant domain movements may occur that enhance the electron transfer to P450s (Supplementary Fig. 1c)[11–14]. The CPR domain movement is considered to apply to the P450BM3-mediated catalysis, in which a "domain swinging" mechanism is proposed to enhance electron transfer from FMN to the heme[15]. On the other hand, it is also suggested that P450BM3 functions as a homodimer[16], which would enable a close approach of cofactors and efficient electron transfer among redox centers from either autologous or neighboring domain[17,18]. In this regard, P450BM3 may employ both intramolecular and intermolecular interactions to transport electrons.

In the other class of self-sufficient P450s, CYP116 P450s contain an N-terminal heme domain fused to a phthalate dioxygenase-type reductase domain that consists of a NAD(P)H reductase domain and a [2Fe-2S] cluster-containing domain (ferredoxin) at the C-terminus (Supplementary Fig. 2a)[19]. The electron transport chain of these P450s is NAD(P)H→FMN→[2Fe-2S]→heme. Several CYP116 P450s have been identified, which display a wide substrate range and catalyze a variety of reactions[20–22]. Notably, the reductase domain of P450 RhF from *Rhodococcus* sp. was artificially fused to various heme domains for the production of high-value chemicals[23–25]. This example highlights the potential of these self-sufficient P450s for biotechnological applications. For CYP116 family, two crystal structures of the heme domain have been reported and their substrate-binding modes were proposed accordingly[26,27]. The complex structures of a heme domain and ferredoxin from P450 systems that acquire electrons from separate redox partners have also been reported[28–31], which demonstrated how the electrons are injected from [2Fe-2S] cluster to the heme (Supplementary Fig. 2b). However, the path for the electrons to travel from the redox partners to the heme group within the polypeptide chain of a self-sufficient P450 remains unclear due to the lack of structural information of the full-length protein. In order to investigate the molecular mechanism of electron transfer of self-sufficient P450, we have solved and herein report the crystal structure of full-length CYP116B46 from *Tepidiphilus thermophilus*[21].

## Results

**Overall structure of full-length CYP116B46.** Recombinant CYP116B46 from *T. thermophilus* was expressed, purified and crystallized. The crystal structure was determined at 2.13-Å resolution (Supplementary Table 1). The electron density map showed a continuous chain from residues 27 to 779, and the corresponding protein model was built unambiguously. The enzyme consists of three domains in the N-to-C-terminus order: the heme domain, the reductase domain and the ferredoxin domain (Fig. 1). The three domains are connected by linker 1 (A437-V464) and linker 2 (E682-R692). The ferredoxin domain is located between the heme domain and the reductase domain, packing against one another (Fig. 1). The interfaces between the domains are mainly constituted by hydrogen bonds and salt bridges and the buried areas are: heme domain and reductase domain, 835.1 Å$^2$; reductase domain and ferredoxin, 619.8 Å$^2$; heme domain and ferredoxin, 580.6 Å$^2$. The binding sites for each of the prosthetic groups, heme, FMN and binuclear Fe-S cluster ([2Fe-2S]), were found in the heme domain, the reductase domain, and the ferredoxin domain, respectively (Fig. 1).

**Structural analysis of the heme domain.** The heme domain is very similar to the recently reported N-terminal region of CYP116B46 (denoted CYP116B46-N), albeit a small structural difference is observed with Cα root-mean-square deviation, Cα

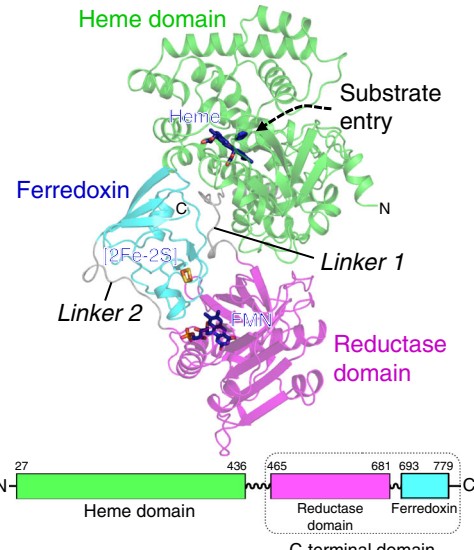

**Fig. 1 Overall structure of CYP116B46.** The polypeptide structure is presented as a cartoon model, with the three domains and connecting linkers in different colors. The heme domain, reductase domain and ferredoxin domain are colored in green, magenta and cyan. The bound ligands that are shown in stick models are labeled, and so are the N- and C-terminus. The lower panel shows the primary structure of CYP116B46 with the amino acid numbers indicated for each region.

RMSD, of 1.154 Å[27] (Supplementary Fig. 3). Differences are mainly observed in helix αB', αE, αF and αG, with helix αF split into αF' and αF" in the CYP116B46-N structure (helix numbering follows ref. [27]) (Supplementary Fig. 3). These regions are distal to the C-terminal domain of the full-length protein and thus the structural variations are likely not related to the presence of the C-terminal domain in our structure (Supplementary Fig. 3a). Notably, the crystal structure of the heme domain from CYP116B5 P450 (denoted as CYP116B5-N) that shares 48% sequence identity with CYP116B46 displays a higher structural similarity to the corresponding region in our full-length structure (Cα RMSD, 0.739 Å) (Supplementary Fig. 3b). In particular, helix αB', αE, αF, and αG in CYP116B5-N and CYP116B46-N adopt various conformations[26]. It is hypothesized that the CYP116B5-N adopts a closed form when a ligand is bound to heme iron. The helices displaying variable conformations mentioned above may move to open a tunnel to the protein surface when the active site becomes vacant[26]. The iron-heme coordination and cysteine-pocket in the P450 domain of the full-length structure are the same as previously reported[27] (Supplementary Fig. 4). Electron density larger than a $H_2O$ molecule was found in the corresponding position of the sixth ligand of the heme Fe (Supplementary Fig. 4). An imidazole molecule was modeled into the electron density map because a high concentration of imidazole was present in the crystallization solution (0.1 M, see Methods). In addition, imidazole bound to the heme iron was also observed in other P450 crystal structures[32–34].

**Structural analyses of the reductase domain.** The reductase domain features two functional parts: an FMN-binding motif and an NADPH-binding motif (Supplementary Fig. 5a). The FMN-binding motif is a six-stranded antiparallel β-barrel while the NADPH-binding motif adopts a Rossmann fold[35] containing a β-sheet comprising five parallel β-strands that is sandwiched by two helices on both sides. The FMN bound in the reductase domain was co-purified with the recombinant protein. Co-crystallization and soaking experiments were carried out in an attempt to obtain an NADPH or NADP+ complex structure, but no electron density was observed for the dinucleotide.

The reductase domain and ferredoxin domain of CYP116B46 and the phthalate dioxygenase reductase from *Pseudomonas cepacis*[36] superimpose very well (Supplementary Fig. 5b), but a slight variation in the domain arrangement is observed (Supplementary Fig. 5c). The conformational variation is not surprising, because CYP116B46 has a heme domain fused directly to the N-terminus of the reductase domain.

**Analysis of the CYP116B46 electron transfer pathway.** The domain arrangement of CYP116B46 displayed in the crystal structure correlates with the direction of electron transport: NADPH→FMN→[2Fe-2S]→heme (Fig. 2a). As mentioned above, the attempt to obtain NADPH or NADP+ complex crystals failed. In order to indicate the NADPH binding site, an NADP+ molecule from the complex structure of ferredoxin-NADP+ reductase was modeled to the reductase domain (Supplementary Fig. 6). First, the substrate-binding pocket formed above the heme group is oriented towards the solvent, allowing substrate entry (Figs. 1 and 2a). The redox equivalents generated by the reductase domain are transported through the ferredoxin domain to the heme domain. In the reductase domain, the *re* side of flavin is facing the putative NAPDH-binding motif and the dimethylbenzene moiety is oriented towards the [2Fe-2S] cluster (Fig. 2a). Notably, the C8 methyl group of FMN is very close to the Fe2 of the binuclear cluster (7.9 Å, Fig. 2a, b), a distance that permits an efficient electron shuttle between the redox centers in

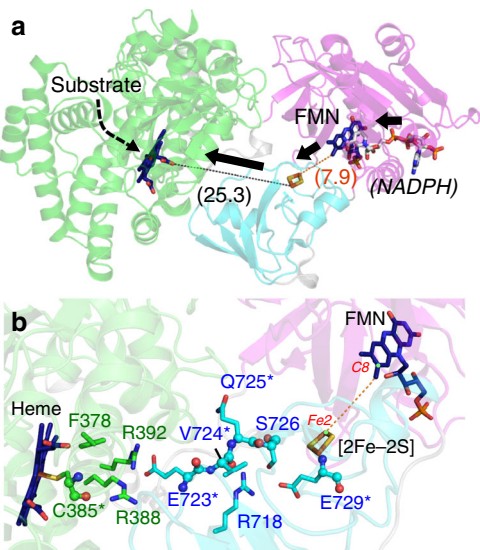

**Fig. 2 Intramolecular electron transfer pathway. a** The CYP116B46 structure is presented as in Fig. 1. The NADPH modeled from pea FNR is displayed as white stick. The thick arrows indicate the direction of electron transfer and the dashed arrow represents the putative portal for substrate entry. Dashed lines connect the redox centers observed in the structure and the corresponding distances (in Å) are also indicated. Black and orange dashed lines represent edge-to-edge distance between heme and [2Fe-2S] cluster and [2Fe-2S] cluster and FMN, respectively. **b** A close-up view of (**a**). The electron transfer path between the C8 methyl group of FMN and the Fe2 of [2Fe-2S] cluster is indicated by a dashed line. The protein residues that may constitute the long-range electron transfer path between [2Fe-2S] and heme are shown as sticks. Residues that may contribute both main chain and side chain to the electron delivery path are noted by asterisks and their main chains are displayed as spheres. Residues colored in green and cyan belong to the heme and ferredoxin domain, respectively. The orange dashed line indicates the distance between FMN C8 methyl group and [2Fe-2S] cluster Fe2 atom.

an iron-sulfur flavoprotein reductase of the phthalate dioxygenase system[36]. In contrast to the close proximity of FMN and [2Fe-2S], the edge-to-edge straight-line distance between the [2Fe-2S] cluster and the heme group is 25.3 Å (Fig. 2a). The distance is too long for efficient electron transfer between the two redox centers, and therefore some amino acid residues or additional co-factors on the path should be involved in relaying the redox equivalents. The tunnel between the [2Fe-2S] and the heme domain is narrow and lacks apparent co-factor-binding features. Therefore, we assume that some of the amino acids along the path may involve the electron transfer. Four residues in the heme domain and six residues in the ferredoxin domain were identified on the path from [2Fe-2S] to the heme (Fig. 2b). These are mainly polar residues and we expect that most of them contribute their side chains to constitute the electron transport path. In addition, main chain of some residues may participate to form the path (E729, S726, Q725 and E723) (Fig. 2b). The hydrophobic residue V724 might contribute to the electron transfer mainly via the main chain and C385 coordinates the heme iron that is invariantly found in P450[37], thus these two residues were not mutated.

The other residues were each mutated to Ala and the decanoic acid hydroxylation rate[38] of all variants was measured. As shown in Fig. 3a, the mutants R388A, R718A, E723A, S726A, and E729A showed lower activity compared with the wild-type enzyme while Ala substitution of R392 and Q725 showed minimal effects on enzyme activity. The decreased enzyme activity should not be due to the protein stability because all recombinant proteins are

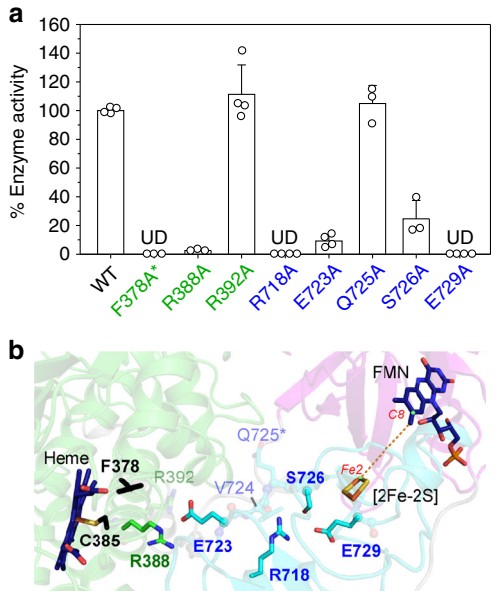

**Fig. 3 Uncovering the hidden route. a** The activity of Ala variants of several residues that are located between [2Fe-2S] and heme was measured. Triplicate or quadruple assay was performed in each independent experiment and the average (mean) and standard error (standard deviation, s.e.m.) for each group were calculated. Three independent experiments were conducted and similar results were yielded. The relative enzyme activity of each variant is presented as a percentage of the wild-type enzyme. The corresponding data points are displayed as dots. The sample size is 4 for WT, R392A, R718A, E723A, and E729A, and is 3 for F378A, R388A, Q725A, and S726A. UD, undetectable. Asterisk indicates that F378A lacked CO-binding capacity. Source data are provided as a Source Data file. The residues that may play a role in enzyme activity are highlighted and shown in (**b**). Residue F378 and C385 that may be required to bind to the heme are colored in black. The labels and color schemes of (**b**) are the same as in Fig. 2b.

resistant to 50 °C treatment (purification procedure, see Methods), and their quality was confirmed by SDS-PAGE and carbon monoxide difference spectroscopy prior to the activity measurement (except for F378A, see below). Notably, the carbon monoxide binding capacity of F378A was undetectable, suggesting that F378 may be involved in heme binding. This indicates that the role of F378 in CYP116B46 is more complex and more experiments are required to address its function. These experimental results demonstrate that residue E729, S276, R718, E723, and R388 that are located between [2Fe-2S] cluster and heme may play a role in CYP116B46 activity (Fig. 3b).

## Discussion

In summary, we report the full-length crystal structure of a self-sufficient P450 that contains a phthalate dioxygenase reductase-type redox partner at the C-terminus. The domain arrangement aligns well with the direction of electron transfer, indicating a plausible pathway from FMN via the [2Fe-2S] cluster to heme. Although NADPH is not observed in our structure, its binding site can be indicated by molecular modeling. The distance between FMN and [2Fe-2S] cluster is estimated to 7.9 Å, which allows effective electron transport[39]. Therefore, our structure should manifest the electron transfer path between FMN and [2Fe-2S] cluster. Compared with the proximity of FMN-to-[2Fe-2S], the edge-to-edge straight-line distance between [2Fe-2S] cluster and heme is estimated to around 25 Å, longer than those observed in complex structures of heme domain and [2Fe-2S]-

containing domain (<20 Å, see Supplementary Fig. 2). In these previous reports, several amino acids that are located between heme and [2Fe-2S] cluster were proposed to provide electrostatic forces to tunnel the electron transfer path[29,30]. Peptide-mediated electron transfer has been considered to play a role in many biological events[40], and the electronic coupling through covalent and hydrogen bonds are believed to be stronger than those across Van der Waals gaps[41]. In the present study, several polar residues that are located between [2Fe-2S] and heme in our structure of CYP116B46 were also found to play a role in catalytic reaction (Fig. 3a). Therefore, we hypothesized that these residues may participate in electron tunneling from [2Fe-2S] cluster to heme.

The long distance between [2Fe-2S] and heme could complicate the electron transport process and result in inefficient enzyme activity if the structural arrangement is adopted during the catalytic reaction. This may explain the lower activity of CYP116 P450s comparing to other P450s (e.g., P450BM3). Nevertheless, CYP116B46 is more efficient than the other similar P450s[21], thus the structural information reported here should serve as an important guidance to explore the important features of CYP116 P450s. On the other hand, our structure may only display the conformations deployed to shuttle electrons from FMN to [2Fe-2S]. It is likely that CYP116B46 also exerts domain movement to bring [2Fe-2S] closer to the heme and deliver electrons as that is proposed for P450BM3[15]. In this context, alternative domain arrangement of CYP116B46 might exist and more functional and structural investigations are required to address these events.

Numerous efforts have been made to engineer the P450 heme domain to overcome many challenging reactions. Engineering the redox partners also holds promises in enhancing the P450 activity, especially for the self-sufficient P450s. In this context, the full-length CYP116B46 structure not only increases our understanding about the molecular mechanism of self-sufficient P450s, but also provides a basis to guide redox partner engineering. Similar strategy has been applied to engineer an alien ferredoxin to support a native-like P450 catalytic activity[42]. Notably, CYP116B46 from *T. thermophilus* is a highly potent and thermostable protein. A more recent report describes its activity in catalyzing region- and enatio-selective C-H-lactonizations of saturated fatty acids[38]. Therefore, CYP116B46 is an attractive subject of research and the structural information revealed in the present work should merit its further applications.

## Methods

**Plasmid construction.** The gene encoding full-length CYP116B46 from *Tepidiphilus thermophilus* (GenBank, WP_055423153.1) was chemically synthesized (Supplementary Table 2) and cloned to the pET46 Ek/LIC vector by Sangon Biotechnology. The plasmid was transformed to *E. coli* BL21 (DE3) (purchased from TransGen Biotech, Beijing, CN) for recombinant protein expression. The plasmids encoding CYP116B46 variants were constructed by using QuickChange site-directed mutagenesis kit (Agilent Technologies, Santa Clara, CA, USA) following the manufacturer's instructions with oligonucleotides listed in Supplementary Table 3. The resulting plasmids were verified by sequencing.

**Recombinant protein expression and purification.** A single transformant of pET46 Ek/LIC expressing wild-type or mutated CYP116B46 gene was grown overnight at 37 °C in LB containing 100 μg mL⁻¹ ampicillin. Ten liters of fresh LB medium with ampicillin (100 μg mL⁻¹) were each inoculated with 200 mL overnight culture and grown to an OD600 of 0.6. The protein expression was induced by 0.4 mM IPTG (isopropyl β-D-1-thiogalactopyranoside) at 16 °C for 18 h. The cells were harvested by centrifugation and then resuspended in 100 mL of lysis buffer containing 20 mM Tris, 150 mM NaCl, and 10 mM imidazole, pH 8.0. Cell disruption was conducted by using a French press (GuangZhou JuNeng Biology and Technology Co. Ltd, Guangzhou, China), and then the lysate was centrifuged at 30,000 × g for 30 min to remove debris. The supernatant was heated at 50 °C for 20 min, and the precipitation was removed by centrifugation at 30,000 × g for 30 min. The extract was loaded onto a Ni-NTA column that was equilibrated by lysis buffer containing 20 mM Tris, 150 mM NaCl, and 10 mM imidazole. The recombinant protein was eluted using a 10–500 mM imidazole gradient. The target

protein-containing fractions were collected and further purified by anion-exchange chromatography using a DEAE column (GE Healthcare). Prior to crystallization trials, protein solutions were concentrated to 60 mg mL$^{-1}$, and stored at −80 °C.

**Protein crystallization and diffraction data collection**. Initial crystallization screening was performed using 768 different reservoir compositions from Hampton Research (Laguna Niguel, California, USA). All of the crystallization experiments were conducted at 25 °C, using the sitting-drop vapor-diffusion method. In general, 1 μL protein (at a concentration of 30 mg mL$^{-1}$) was mixed with 1 μL reservoir solution in 96-well Cryschem plates (Hampton Research) and equilibrated against 100 μL reservoir solution. Initial crystals of CYP116B46 were obtained within 4 days using PEGRxTM screen 47 (12% polyethylene glycol 20,000, 0.1 M imidazole, pH 7.0). The crystallization condition was then optimized to 9% polyethylene glycol 20,000, 0.1 M imidazole, pH 7.0. Prior to the data collection, crystals were soaked in a cryo-protectant containing 10% polyethylene glycol 20,000, 0.1 M imidazole, 25% ethylene glycol, pH 7.0. The X-ray diffraction datasets were collected at beam line TPS 05A of the National Synchrotron Radiation Research Center (NSRRC, Hsinchu, Taiwan). Data was processed by using HKL2000[43].

**Structure determination and refinement**. The crystal structure of CYP116B46 was solved by molecular replacement with the Phaser program[44] using CYP116B46 heme domain (PDB ID, 6GII [https://doi.org/10.2210/pdb6GII/pdb])[27] and phthalate dioxygenase reductase from *Pseudomonas cepacia* (PDB ID, 2PIA [https://doi.org/10.2210/pdb2PIA/pdb])[36] as search models. Prior to structure refinement, 5% randomly selected reflections were set aside for calculating R$_{free}$ as a monitor of model quality. Model adjustment and refinement were carried out by using Refmac5[45] and Coot[46]. All graphics for the protein structures were prepared by using the PyMOL program (http://pymol.sourceforge.net/). Data collection and refinement statistics are summarized in Supplementary Table 1.

**Enzyme activity measurement**. The enzyme activity of CYP116B46 was estimated by decanoic acid hydroxylation assay as previously described[38]. The reaction was carried out in 1 mL of 100 mM phosphate buffer (pH 8.0) at 30 °C and at 240 × g in the presence of 10 μM enzyme, 0.4 mM decanoic acid, 1 mM NADP$^{+}$, 2 U glucose dehydrogenase and 50 mg mL$^{-1}$ glucose. After 5 h, an equal volume of ethyl acetate was added to the reaction mixture to quench the reaction. The resulting solution was centrifuged at 13,000 × g for 2 min, and the organic phase was dried over Na$_2$SO$_4$ overnight. The resulting solid was dissolved in 30 μL N-methyl-N-(trimethylsilyl) trifluoroacetamide (MSTFA) and 60 μL pyridine. The derivatization reaction was performed at 65 °C for 1 h and then the mixtures were used for GC analysis to determine the product formation. Reaction was performed in triplicate, and the total turnover number (TTN) was calculated based on the ratio of the amount of product formed to the amount of enzyme used.

**GC analysis**. GC analysis was performed with an SH-Rtx-1 column. 90 μL EtOAc containing an internal standard (25 mM n-decane) was added into derivative mixture. Afterwards, the samples were analyzed using a SHIMADZU Nexis GC-2030 system equipped with a FID detector and SH-Rtx-1 column (30 m × 25 mm, 0.25 μm). The temperatures of injector and detector were 250 °C and 280 °C, respectively. The temperature program was as follows: 5 °C per min from 50 °C to 120 °C, 40 °C per min to 240 °C, and held at 240 °C for 1 min.

**Reporting summary**. Further information on research design is available in the Nature Research Reporting Summary linked to this article.

## Data availability

The atomic coordinates and structure factors have been deposited in the Protein Data Bank under accession code 6LAA [https://doi.org/10.2210/pdb6laa/pdb]. The source data underlying Fig. 3a is provided as a Source Data file. Other data are available from the corresponding authors upon reasonable request.

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

## Acknowledgements

This work has been supported by the National Key Research and Development Program of China (2019YFA070030), the National Natural Science Foundation of China (31870790 and 31971205), the Natural Science Foundation of Tianjin (18JCYBJC24300), and China Postdoctoral Science Foundation (2019M662572). We thank NSRRC (National Synchrotron Radiation Research Center, Taiwan) for access to beam line TPS-05A that contributed for the synchrotron data collection.

## Author contributions

L.Z., Z.X., S.Z., X.L., Y.H., and X.Y. carried out cloning, mutagenesis, protein purification and crystallization. Z.L., S.Z., and J.M. measured enzyme activity. W.L., J.W.H. and R.T.G. collected crystallographic data. T.P.K., R.T.G. and C.C.C. solved, refined, and analyzed the crystal structure. L.M., R.T.G. and C.C.C designed mutants, supervised researches and prepared the manuscript with contributions from all authors.

## Competing interests

The authors declare no competing interests.
