## [Peer Review File · Nature Communications]

Reviewers' comments:

Reviewer #1 (Remarks to the Author):

This is an outstanding work since it reports the first crystal structure of a self-sufficient cytochrome P450 in the full-length version. Many efforts have been made in the field with many unsuccessful attempts to crystallize for example cytochrome P450 BM3.

For the CYP116 family, to obtain the full-length protein in a holo form, suitable for crystallization, is already a challenge. Here, the crystallographic data are strong at a very good resolution. Moreover, the activity on a fatty acid substrate is reported. Thus, the manuscript is potentially suitable for publication in Nature Communications.

However, there are some major issues that Authors should address before publication:

1. An electron density in the active site shows the presence of a ligand that directly coordinates the heme iron. The presence of oxygen or hydrogen peroxide ion requires reduction of the heme iron. So, I wonder if the reductase domain was in a semi-reduced form before crystallization and therefore electron transfer has eventually occurred. Secondly, from where hydrogen peroxide comes from? May be something has happened during data collection?
2. In the crystallization cocktail, imidazole is present. Imidazole is a strong P450 inhibitor that is known to directly coordinate heme iron. Is it possible that the electron density is due to the presence of this compound with a low occupancy or in different orientations? Of course, this is only a suggestion. In any case, the presence of a peroxide ion (if justified) can be highly interesting since the heme domain of CYP116B5 showed a peroxygenase activity when isolated (Ciaramella et al., 2020).
3. An important part of the literature has been ignored by the Authors also in the interpretation of their data. It is nowadays known that electron transfer occurs through large conformational changes that allow domain movements in cytochromes P450. This has been shown for P450 BM3 and for mammalian CPR (see for example DOI: 10.1021/jacs.7b00663; DOI: 10.1016/j.str.2013.06.022). The fact that the Authors find the Fe-S cluster too far from heme for an efficient electron transfer is not surprising, since the P450 community has accepted the idea of a large domain flexibility in self-sufficient cytochromes P450.

I suggest not to put emphasis to the electron transfer route since the crystallographic data themselves are strong enough. A better comparison with the crystal structure already available is missing. For example the fact that the helix F is folded in this crystal structure whereas it is unfolded in the one reported before for the heme domain could suggest some role of the reductase domain?

Minor points:

- Abstract section. It is not clear what the Authors mean with "Ease of gene manipulation". I don't see any gene manipulation, but just a standard gene cloning in the methods section.
- Introduction section. The Authors should mention both the only two crystal structures available for the heme domain of CYP116 family (please include CYP116B5).
- Line 38, remove "the"

Reviewer #2 (Remarks to the Author):

The authors have solved the first crystal structure of a full length catalytically self-sufficient P450. The crystallography and mutational analysis are all fine. However, the conclusions the author's draw on the electron transfer (ET) pathway and the involvement of various residues between the heme and iron-sulfur cluster is very likely wrong. From what we now know about how redox partners dock to P450s, the FMN or FeS redox partner docks on the proximal side of the P450 so that the ET distance is on the order 15Å. P450BM3 also is a catalytically self-sufficient complex and the structure of the FMN domain docked to the heme domain shows the FMN domain docked at the same site found for P450cam. The current and probably correct view on how this works with

P450BM3 and other P450s that use a FMN/FAD reductase is that the FMN directly interacts with the FAD to pick up electrons and then undergoes a large change to swing around and dock to the heme domain. It is very likely that something similar is going on here. What the authors most likely have captured is the structure for FAD-to-FeS ET but not FeS-to-heme ET. The authors need to rethink all of this and give more serious consideration to the "swinging domain" model and/or provide a better justification for their proposed ET mechanism.

Reviewer #3 (Remarks to the Author):

This manuscript reports the full length crystal structure of the self-sufficient P450 fusion protein CYP116B46. This is the first full length structure of a P450 of this type. As such the results and structure should be of interest for those in the field. My concern with the manuscript as it stands is that it is very short and provides a weak overview of electron transfer in P450 systems and therefore how this structure can be interpreted within the context of previous work on self-sufficient and non-self-sufficient P450 systems (see below). There is also lack of biochemical data for the system. That reported by the authors here for the mutants in this manuscript could be presented more clearly.

The results are interesting in that there is a very short FMN to 2Fe2S distance (7.9 angstroms) but the distance between the 2Fe2S cofactor and the heme is much longer (25.3 angstroms).

Self-sufficient p450s, as stated by the authors, are biotechnologically useful as electron transfer partners do not have to be obtained. P450Bm3 has found widespread use as it is very active and efficient. This is often stated to be due to its self-sufficient fused nature but this is not true. It is efficient as it functions as a dimer enabling close approach of cofactors and fast electron transfer steps.

References (P450Bm3 dimer)

DOI: 10.1021/bi701031r

DOI: 10.1074/jbc.RA117.000600

DOI:10.1016/j.febslet.2005.09.023

For the CYP116 family of enzymes the activity rates are as far as I am aware is always low compared to other P450s. These are even lower than those of certain non self-sufficient P450s such as P450cam, which the authors make reference to. This and other similar enzymes have $k_{cat} > 35 \text{ s}^{-1}$ which is much faster than has been reported for the CYP116 family.

This structure could provide an explanation of why and this should be discussed/highlighted in the manuscript. The low activity of the CYP116 enzyme seems to be due to the large distance slow between the 2Fe2S ferredoxin and the heme which would lead to slow electron transfer. Have the authors measured the rate of electron transfer between the different electron transfer sites (it is suggested that they have line 143-145 for the FMN to 2Fe2S transfer but what is this data – it hasn't been shown)? If they haven't have others? If so this should be referenced.

Line 65 - 70 The authors reference structure of Pdx and P450cam (X-ray and NMR). These however are not the only ones available. See the references below for another bacterial example and adrenodoxin and a mitochondrial P450. These all show close approach of the 2Fe2S cluster and the heme centre.

DOI:10.1021/jacs.7b11056

DOI:10.1073/pnas.1019441108

When the authors state that the electron pathways are not known for P450s this is not strictly true for all the examples above as the ferredoxin binds at the proximal face at the closest position it can get to the cysteinate face of the heme. The distance for the electron in these systems and the P450cam system is much shorter than that described here accounting for the higher activity (at least for the bacterial systems). The complex pathway through the many residues proposed by the authors here (lines 149-151) would explain low rates in CYP116 enzymes.

Line 168 It should be noted that engineering of redox partners has already been undertaken in other systems.

Reference

DOI:10.1039/c2cc35968e

Figure 3a The activity measurement are based on decanoic acid oxidation? Is this based on NAD(P)H oxidation or product formation. This seems sound but can the authors provide information on the stability and expression levels of these mutants? Are they able to measure separate rates of electron transfer between the cofactors?

Other comments

Line 98-102 – the identification of the species bound to the heme iron as O₂ is speculative and more information should be provided (or this should be removed). Information on the occupancy and other analysis would be required to validate this.

Imidazole is used in the crystallization buffer. Does this bind to the P450 and cause a shift in the heme Soret band?

Line104. Do the authors need a reference to the Rossmann fold?

Line 136 the authors need to explain why V724 is thought to be critical to the electron transfer and why it was not mutated?

Overall a very interesting and important structure but a more thorough evaluation of what it reveals is required. If the authors could expand on the significance of their results in the context of what is already known about P450 electron transfer they would have a very nice manuscript.

Reviewers' comments:

Reviewer #1 (Remarks to the Author):

This is an outstanding work since it reports the first crystal structure of a self-sufficient cytochrome P450 in the full-length version. Many efforts have been made in the field with many unsuccessful attempts to crystallize for example cytochrome P450 BM3. For the CYP116 family, to obtain the full-length protein in a holo form, suitable for crystallization, is already a challenge. Here, the crystallographic data are strong at a very good resolution. Moreover, the activity on a fatty acid substrate is reported. Thus, the manuscript is potentially suitable for publication in Nature Communications.

Response: We thank the Reviewer's efforts in reviewing this manuscript and are also very grateful for the very helpful and constructive suggestions.

However, there are some major issues that Authors should address before publication:

1. An electron density in the active site shows the presence of a ligand that directly coordinates the heme iron. The presence of oxygen or hydrogen peroxide ion requires reduction of the heme iron. So, I wonder if the reductase domain was in a semi-reduced form before crystallization and therefore electron transfer has eventually occurred. Secondly, from where hydrogen peroxide comes from? May be something has happened during data collection? In the crystallization cocktail, imidazole is present. Imidazole is a strong P450 inhibitor that is known to directly coordinate heme iron. Is it possible that the electron density is due to the presence of this compound with a low occupancy or in different orientations? Of course, this is only a suggestion. In any case, the presence of a peroxide ion (if justified) can be highly interesting since the heme domain of CYP116B5 showed a peroxygenase activity when isolated (Ciaramella et al., 2020).

Response: We thank the Reviewer for raising the question about the sixth ligand of heme iron observed in the structure. As the Reviewer mentioned, the presence of an O₂ or H₂O₂ in the active site implies that either the electron transfer took place in the crystallized protein or the heme domain is per se a peroxygenase and can bind a peroxide ion. The first possibility is less likely as the electron transfer should occur in the presence of the electron donor,

NADPH for the case of CYP116B46. Our activity measurement indicated that the purified recombinant proteins do not exhibit catalytic activity in the absence of NADPH. The second possibility that was proposed in the references provided by the Reviewer is very interesting, but the peroxygenase activity of CYP116B46 (the protein reported in the present paper) remains to be determined. We will examine the peroxygenase activity of CYP116B46 in the following experiments. Before more experimental results are available, we decided to accept the Reviewer's suggestion and model the map with an imidazole. The related statements are modified to “**Electron density larger than a H₂O molecule was found in the corresponding position of the sixth ligand of the heme Fe (Supplementary Figure 4). An imidazole molecule was modeled into the electron density map because a high concentration of imidazole was present in the crystallization solution (0.1 M, see Methods). In addition, imidazole bound to the heme iron was also observed in other P450 crystal structures³²⁻³⁴.**”.

2. An important part of the literature has been ignored by the Authors also in the interpretation of their data. It is nowadays known that electron transfer occurs through large conformational changes that allow domain movements in cytochromes P450. This has been shown for P450 BM3 and for mammalian CPR (see for example DOI: 10.1021/jacs.7b00663; DOI: 10.1016/j.str.2013.06.022). The fact that the Authors find the Fe-S cluster too far from heme for an efficient electron transfer is not surprising, since the P450 community has accepted the idea of a large domain flexibility in self-sufficient cytochromes P450. I suggest not to put emphasis to the electron transfer route since the crystallographic data themselves are strong enough.

Response: We appreciate the Reviewer's approval on the quality of our structural works and the suggestions on our proposed electron transfer path. The short distance between FMN and [2Fe-2S] displayed in our structure may indicate an efficient electron traveling path. On the other hand, the distance between [2Fe-2S] and heme is estimated around 25 Å, longer than that was observed in complex structures of P450 and FMN- or [2Fe-2S]-containing domain (<20 Å). We agree with the Reviewer that domain movement of CYP116B46 might take place to reduce the distance between [2Fe-2S] and the heme to enhance electron transfer rate. Therefore, we modified our interpretations about the results of mutagenesis experiments, and state that “**These experimental results demonstrate that residue E729, S276, R718, E723,**

and R388 that are located between [2Fe-2S] cluster and heme may play a role in CYP116B46 activity (**Fig. 3b**).". The descriptions related to the electron transport hypothesis have also been modified. We also state that our structure may only display the conformation that facilitate electron transfer from FMN to [2Fe-2S] and further investigations are required to address this issue. These descriptions are included in the first and second paragraph of Discussion section.

Meanwhile, we also describe the conformational change theory of other P450 systems to provide a comprehensive overview of P450 mechanism of action. Structural studies of mammalian CPRs indicate that FMN-binding domain extends to adopt an "open-form" to expose FMN to approach the heme, such that the electrons can be efficiently delivered. This domain swing mechanism is proposed to be applied by P450BM3 because it fuses a homologous CPR at C-terminus. The complex structure of FMN-containing flavodoxin domain and the heme domain of P450BM3 displays a conformation in which FMN orients towards heme proximal side. In such conformation, the flavodoxin domain should be part of the "open-form" of CPRs. These information has been described in the second paragraph of the Introduction section. Furthermore, we also draw a supplementary figure (**Supplementary Figure 1**) to illustrate the domain arrangement of P450BM3.

3. A better comparison with the crystal structure already available is missing. For example the fact that the helix F is folded in this crystal structure whereas it is unfolded in the one reported before for the heme domain could suggest some role of the reductase domain?

Response: We thank the Reviewer for asking the question regarding the various conformation of helix F revealed in a previously reported heme domain structure (CYP116B46-N) and the present full-length structure. The conformational alterations are mainly observed in helices $\alpha B'$, αE , αF and αG , with helix αF split into $\alpha F'$ and $\alpha F''$ in CYP116B46-N. As mentioned in the original version of manuscript, the structural variation is presumed not directly related to the presence of the C-terminal domain as these structural elements are distal to the C-terminus. Instead, we hypothesize that the heme domain may adopt open and closed form by altering these active-site covering helices. Another heme domain structure of CYP116B5, which shares 48% sequence identity to CYP116B46 heme domain, shows a similar conformation to that of the full-length structure (Ciaramella, *Int J. Biol. Macromol* (2019) p.

577). The authors conducted MD simulations and suggested that the conformation they observed may be a closed form while the CYP116B46-N with a broken helix α F being an open form. This interesting hypothesis may be applied to our recent findings, which indicate that the CYP116B46 heme domain adopt a closed form. These structural analyses have been added to the second paragraph of the Result section in the revised manuscript.

Minor points:

- Abstract section. It is not clear what the Authors mean with “Ease of gene manipulation”. I don’t see any gene manipulation, but just a standard gene cloning in the methods section.

Response: We intended to state that the structural information revealed by this work should benefit biotechnological applications of self-sufficient P450s because it is easier to conduct gene manipulation on a single polypeptide chain comparing to those contains multiple gene fragments. We apologize for the confusing descriptions and have modified the abstract. The last sentence in the abstract now reads “These findings not only provide insights into the intramolecular electron transfer of self-sufficient P450s, but also are of interest for biotechnological applications of self-sufficient P450s .”.

- Introduction section. The Authors should mention both the only two crystal structures available for the heme domain of CYP116 family (please include CYP116B5).

Response: These references have been added to the 3rd paragraph of the Introduction section in compliance with the Reviewer’s suggestion. In addition, the structural comparisons of CYP116 family members are demonstrated in the second paragraph of the Result section (see above).

- Line 38, remove “the”

Response: Corrected.

Reviewer #2 (Remarks to the Author):

The authors have solved the first crystal structure of a full length catalytically self-sufficient P450. The crystallography and mutational analysis are all fine. However, the conclusions the author's draw on the electron transfer (ET) pathway and the involvement of various residues between the heme and iron-sulfur cluster is very likely wrong. From what we now know about how redox partners dock to P450s, the FMN or FeS redox partner docks on the proximal side of the P450 so that the ET distance is on the order 15Å. P450BM3 also is a catalytically self-sufficient complex and the structure of the FMN domain docked to the heme domain shows the FMN domain docked at the same site found for P450cam. The current and probably correct view on how this works with P450BM3 and other P450s that use a FMN/FAD reductase is that the FMN directly interacts with the FAD to pick up electrons and then undergoes a large change to swing around and dock to the heme domain. It is very likely that

something similar is going on here. What the authors most likely have captured is the structure for FAD-to-FeS ET but not FeS-to-heme ET. The authors need to rethink all of this and give more serious consideration to the "swinging domain" model and/or provide a better justification for their proposed ET mechanism.

Response: We thank the Reviewer for commenting our works. We agree with the Reviewer that our structure of CYP116B46 may only display the conformation that is deployed for electron shuttling from FMN to [2Fe-2S] cluster. The long distance between [2Fe-2S] and heme might not merit the direct electron transport, thus further conformational change is possible. Therefore, we revised the manuscript to make these points clearer:

- (1) The currently acknowledged mechanisms of electron transport of two self-sufficient P450s are described in the second and third paragraph of the Introduction section. In the second paragraph, we describe the complex structure of flavodoxin and the heme domain of P450BM3. This structure indicates that FMN is located proximal to the heme, with a distance around 18 Å (**Supplementary Figure 1b**). Furthermore, structures of mammalian CPRs that adopt closed and open conformation are also demonstrated (**Supplementary Figure 1c**). From these results, P450BM3 that fuses a homologous CPR domain on the C-terminus should exert FMN-containing flavodoxin movement to enhance the electron

transport rate.

- (2) In the third paragraph of Introduction, the complex structures of P450 and a partial domain of redox partner that contains [2Fe-2S] cluster are described. As shown in Supplementary Figure 2b, the distance between two cofactors are around 13 Å.
- (3) We modified our interpretations about the findings of mutagenesis experiments, and state that “These experimental results demonstrate that residue E729, S276, R718, E723, and R388 that are located between [2Fe-2S] cluster and heme may play a role in CYP116B46 activity (**Fig. 3b**)”. The descriptions related to the electron transport hypothesis have also been modified. We also state that our structure may only display the conformation that facilitate electron transfer from FMN to [2Fe-2S] and further investigations are required to address this issue. These descriptions are included in the first and second paragraph of Discussion section. In particular, we noted that a domain movement similar to that occur in P450BM3 is likely a measure for CYP116B46 to deliver the electrons. “On the other hand, our structure may only display the conformations deployed to shuttle electrons from FMN to [2Fe-2S]. It is likely that CYP116B46 also exerts domain movement to bring [2Fe-2S] closer to the heme and deliver electrons as that is proposed for P450BM3¹⁵. In this context, alternative domain arrangement of CYP116B46 might exist and more functional and structural investigations are required to address these events.”

Reviewer #3 (Remarks to the Author):

This manuscript reports the full length crystal structure of the self-sufficient P450 fusion protein CYP116B46. This is the first full length structure of a P450 of this type. As such the results and structure should be of interest for those in the field. My concern with the manuscript as it stands is that is very short and provides a weak overview of electron transfer in P450 systems and therefore how this structure can be interpreted within the context of previous work on self-sufficient and non-self-sufficient P450 systems (see below). There is also lack of biochemical data for the system. That reported by the authors here for the mutants in this manuscript could be presented more clearly.

Responses: We thank the Reviewer for the comments and have revise the manuscript accordingly.

1. The results are interesting in that there is a very short FMN to 2Fe2S distance (7.9 angstroms) but the distance between the 2Fe2S cofactor and the heme in much longer (25.3 angstroms). Self-sufficient p450s, as stated by the authors, are biotechnologically useful as electron transfer partners do not have to be obtained. P450Bm3 has found widespread use as it is very active and efficient. This is often stated to be due to its self-sufficient fused nature but this is not true. It is efficient as it functions as a dimer enabling close approach of cofactors and fast electron transfer steps.

Response: We thank the Reviewer for providing these very important information that indicate the intramolecular and intermolecular electron transfer in P450BM3. These references and related descriptions have been added to the second paragraph of the Introduction section, which read “On the other hand, it is also suggested that P450BM3 functions as a homodimer¹⁶, which would enable a close approach of cofactors and efficient electron transfer among redox centers from either autologous or neighboring domain^{17,18}. In this regard, P450BM3 may employ both intramolecular and intermolecular interactions to transport electrons.”

References:

16. Neeli, R. et al. The dimeric form of flavocytochrome P450 BM3 is catalytically functional as a fatty acid hydroxylase. *FEBS Lett* **579**, 5582-5588 (2005).
17. Kitazume, T., Haines, D.C., Estabrook, R.W., Chen, B. & Peterson, J.A. Obligatory intermolecular electron-transfer from FAD to FMN in dimeric

P450BM-3. *Biochemistry* **46**, 11892-11901 (2007).

18. Zhang, H. et al. The full-length cytochrome P450 enzyme CYP102A1 dimerizes at its reductase domains and has flexible heme domains for efficient catalysis. *J Biol Chem* **293**, 7727-7736 (2018).

2. For the CYP116 family of enzymes the activity rates are as far as I am aware is always low compared to other P450s. These are even lower than those of certain non self-sufficient P450s such as P450cam, which the authors make reference to. This and other similar enzyme have $k_{cat} > 35 \text{ s}^{-1}$ which is much faster than has been reported for the CYP116 family. This structure could provide an explanation of why and this should be discussed/highlighted in the manuscript. The low activity of the CYP116 enzyme seems to be due to the large distance slow between the 2Fe2S ferredoxin and the heme which would lead to slow electron transfer.

Response: We thank the Reviewer for the comment and agree that the long distance between [2Fe-2S] cluster and heme observed in the present structure of full-length CYP116B46 may provide an explanation for the lower catalytic activity of this type of P450s. These descriptions have been added to the second paragraph of the Discussion section, which read “The long distance between [2Fe-2S] and heme could complicate the electron transport process and result in inefficient enzyme activity if the structural arrangement is adopted during the catalytic reaction. This may explain the lower activity of CYP116 P450s comparing to other P450s (e.g. P450BM3). Nevertheless, CYP116B46 is more potent than the other similar P450s²¹, thus the structural information reported here should serve as an important guidance to explore the important features of CYP116 P450s.”

References:

21. Tavanti, M., Porter, J.L., Sabatini, S., Turner, N.J. & Flitsch, S.L. Panel of new thermostable CYP116B self-sufficient cytochrome P450 monooxygenases that catalyze C–H activation with a diverse substrate scope. *ChemCatChem* **10**, 1042-1051 (2018).

3. Have the authors measured the rate of electron transfer between the different electron transfer sites (it is suggested that they have line 143-145 for the FMN to 2Fe2S transfer but what is this data – it hasn't been shown)? If they haven't have others? If so this should be referenced.

Response: The Reviewer raised a very important question that regards the

electron transfer rate between each redox centers. We agree with the Reviewer that it is very important to measure the electron transfer of full-length P450s. Unfortunately, we were not able to conduct these experiments. We will try to see if we can establish protocols and instruments to conduct these assays. We believe these investigations should yield many important results to illustrate the P450 mechanism, and these finding should go to follow-up papers.

4. Line 65 - 70 The authors reference structure of Pdx and P450cam (X-ray and NMR). These however are not the only ones available. See the references below for another bacterial example and adrenodoxin and a mitochondrial P450. These all show close approach of the 2Fe₂S cluster and the heme centre.

Response: We thank the Reviewer for this information and have added the referred references and related descriptions to the 3rd paragraph of the Introduction section to give an overall view of current understanding of electron transfer routes between heme and 2[Fe-S] cluster. In addition, we also drew a supplementray figure to illustrate the domain arrangement of CYP116 P450s and display the complex structures of P450 and various ferredoxins (Supplementary Figure 2). These statements should provide an overall picture of current knowledge of the topology of CYP116 P450s, which read “For CYP116 family, two crystal structures of the heme domain have been reported and their substrate-binding modes were proposed accordingly^{26,27}. The complex structures of a heme domain and ferredoxin from P450 systems that acquire electrons from separate redox partners have also been reported²⁸⁻³¹, which demonstrated how the electrons are injected from [2Fe-2S] cluster to the heme (**Supplementary Figure 2b**).”

Supplementary Figure 2 Domain organization of CYP116 P450s. (a) The overall domain arrangement of CYP116 P450s. The phthalate dioxygenase-type reductase domain fused to the C-terminus of heme domain is indicated by dash line. (b) Complex structures of (left and middle) P450cam heme domain and putidaredoxin^{5, 6}, and (right) CYP11A1 heme domain and adrenodoxin⁷. Protein structures are presented in cartoon model with their PDB ID indicated below. The heme domains and redox transfer domains are colored in green and magenta. Co-factors bound in each domain are displayed as sticks. The edge-to-edge straight-line distance cofactors within each complexes are shown in parentheses (unit, Å).

- When the authors state that the electron pathways are not known for P450s this is not strictly true for all the examples above as the ferredoxin binds at the proximal face at the closest position it can get to the cysteinate face of the heme. The distance for the electron in these systems and the P450cam system is much shorter than that described here accounting for the higher activity (at least for the bacterial systems). The complex pathway through the many residues proposed by the authors here (lines 149-151) would explain low rates in CYP116 enzymes.

Response: We thank the Reviewer for the comments regarding to the

electron transfer mechanism of P450s. We included more references to describe the current understanding of the electron transfer routes in two main types of self-sufficient P450s: P450BM3 and CYP116. These information are included in the second and third paragraph of the Introduction section, respectively. We also agree with the Reviewer that our structure may, at least in part, explain the low efficacy of CYP116 P450s, and added related discussions in the revised manuscript (see comment 2)

Line 168 It should be noted that engineering of redox partners has already been undertaken in other systems. Reference DOI:10.1039/c2cc35968e

Response: We appreciate the Reviewer's suggestion and have added a sentence to note the reference in the second paragraph of the Discussion section in the revised manuscript. The sentence reads "Similar strategy has been applied to engineer an alien ferredoxin to support a native-like P450 catalytic activity⁴²".

6. Figure 3a The activity measurement are based on decanoic acid oxidation? Is this based on NAD(P)H oxidation or product formation. This seems sound but can the authors provide information on the stability and expression levels of these mutants? Are they able to measure separate rates of electron transfer between the cofactors?

Response: Yes the mutagenesis experiment was conducted based on decanoic acid oxidation. We apologize for the arbitrary descriptions in the original version of manuscript, and have made clearer statements in the revised manuscript. The first sentence in the second paragraph of the last part of the Result section: "The other residues were each mutated to Ala and the decanoic acid hydroxylation rate³⁸ of all variants was measured."

These assays were conducted by using purified recombinant protein. The recombinant proteins of each variants were expressed and purified in *E. coli* following the same protocol as that used to obtain wild type protein. Prior to the activity measurement, the quality and quantity of all proteins were validated in several ways: (1) SDS-PAGE for protein purity; (2) 280 nm absorbance for total protein concentration; (3) functional P450 concentration measured by carbon monoxide difference spectroscopy. Notably, all of the recombinant proteins were treated at 50 °C for 20 minutes during the purification, suggesting that the thermostability is preserved in all of the variant proteins. Regarding to the separate rates of electron transfer between

the cofactors, we are not able to do these experiment at this stage. We agree with the Reviewer that it is essential to address the electron transfer route of full-length P450s and will establish protocols and instruments to conduct these assays. The results of these further investigations should go to the other paper.

Other comments

7. Line 98-102 – the identification of the species bound to the heme iron as O₂ is speculative and more information should be provided (or this should be removed). Information on the occupancy and other analysis would be required to validate this. Imidazole is used in the crystallization buffer. Does this bind to the P450 and cause a shift in the heme Soret band?

Response: Indeed we do not have experimental evidence to validate the existence of oxygen (or other related species) bound to the heme iron. Instead, we agree with the Reviewer that it is more likely that an imidazole should be what was observed. In addition, binding of an imidazole to the heme iron is often observed when we solved crystal structures of heme domain of other types of P450. Therefore, we revise our structure and model the map bound to the heme iron with an imidazole in compliance with the Reviewer's suggestion.

8. Line104. Do the authors need a reference to the Rossmann fold?

Response: A reference to review the structure and function of Rossmann fold has been added to the referred sentence.

9. Line 136 the authors need to explain why V724 is thought to be critical to the electron transfer and why it was not mutated?

Response: We thank the Reviewer for asking this question. As we mentioned in the manuscript, V724 was found to locate on the putative path of electron transfer route. Nevertheless, this hydrophobic residue may contribute its main chain O and N atoms during the electron relay. We did not construct Ala variant of V724 because we expected no change will be seen for the main chain architecture. In addition to V724, main chain of some other residues are found to be on the track. To make this point clearer, the main chain of V724 along with other residues is displayed on the Figure 2b in the revised manuscript. Related descriptions are also added to the end of the first paragraph of the last part of the Result section: “**These are mainly polar residues and we expect that most of them contribute their side chains to constitute the electron transport**”

path. In addition, main chain of some residues may also participate to form the path (E729, S726, Q725 and E723) (**Fig. 2b**). The hydrophobic residue V724 might contribute to the electron transfer mainly via the main chain and C385 coordinates the heme iron that is invariantly found in P450³⁷, thus these two residue were not mutated.”.

10. Overall a very interesting and important structure but a more thorough evaluation of what it reveals is required. If the authors could expand on the significance of their results in the context of what is already known about P450 electron transfer they would have a very nice manuscript.

Response: We thank the Reviewer for the positive comments. We also appreciate the Reviewer’s suggestions and have incorporated this constructive information to the revised manuscript.

REVIEWERS' COMMENTS:

Reviewer #1 (Remarks to the Author):

The Authors have addressed all the points and changed the manuscript accordingly.

I recommend acceptance but there are still a couple of minor corrections:

1. Line 231-232: I don't understand what the sentence "CYP116B46 is more potent than the other similar P450s" means. Do the Authors mean more efficient?

2. Supplementary Figure 4. I don't understand why the electron density map of heme with water is shown in the bottom panels. I think it can be removed because it is misleading.

Also, letters A, B, etc.... should be added to all the figures rather than top, middle and bottom (Suppl Fig 3 and 4).

Reviewer #3 (Remarks to the Author):

The authors have done a good job adjusting the manuscript to reflect all the reviewers' comments.

It could do with a quick check of the language before publication. Some may quibble with the details and interpretation of the results based on these structures but it is for the moment the best structure available for these enzymes. Therefore it is in my opinion an important piece of research that has been well performed and should be published.

Minor corrections.

Page 7 last sentence - these two residues were not mutated

Page 9 paragraph 1 Van der Waals

REVIEWERS' COMMENTS:

Reviewer #1 (Remarks to the Author):

The Authors have addressed all the points and changed the manuscript accordingly.

I recommend acceptance but there are still a couple of minor corrections:

1. Line 231-232: I don't understand what the sentence "CYP116B46 is more 232 potent than the other similar P450s" means. Do the Authors mean more efficient?

Response: Yes, this sentence means that CYP116B46 is more efficient. We apologize for the confusing expressions and have revised this sentence, which now reads "CYP116B46 is more efficient than the other similar P450s."

2. Supplementary Figure 4. I don't understand why the electron density map of heme with water is shown in the bottom panels. I think it can be removed because it is misleading.

Response: We agree with the Reviewer and the referred image has been removed from Supplementary Figure 4.

Also, letters A, B, etc.... should be added to all the figures rather than top, middle and bottom (Suppl Fig 3 and 4).

Response: We appreciate the Reviewer's advice and have revised the figures as suggested.

Reviewer #3 (Remarks to the Author):

The authors have done a good job adjusting the manuscript to reflect all the reviewers comments.

It could do with a quick check of the language before publication. Some may quibble with the details and interpretation of the results based on this structures but it is for the moment the best structure available for these enzymes. Therefore it is in my opinion an important piece of research that has been well performed and should be published.

Response: We thank the Reviewer for approving our works and also appreciate the Reviewer's efforts in revising the manuscript.

Minor corrections.

Page 7 last sentence - these two residues were not mutated

Response: corrected.

Page 9 paragraph 1 Van der Waals

Response: corrected.